# Dysfunction of Mitochondria in Alzheimer’s Disease: ANT and VDAC Interact with Toxic Proteins and Aid to Determine the Fate of Brain Cells

**DOI:** 10.3390/ijms23147722

**Published:** 2022-07-13

**Authors:** Anna Atlante, Daniela Valenti, Valentina Latina, Giuseppina Amadoro

**Affiliations:** 1Institute of Biomembranes, Bioenergetics and Molecular Biotechnologies (IBIOM)-CNR, Via G. Amendola122/O, 70126 Bari, Italy; d.valenti@ibiom.cnr.it; 2European Brain Research Institute (EBRI), Viale Regina Elena 295, 00161 Rome, Italy; valentina.latina80@gmail.com; 3Institute of Translational Pharmacology (IFT)-CNR, Via Fosso del Cavaliere 100, 00133 Rome, Italy

**Keywords:** Alzheimer’s disease, mitochondria, amyloid β-peptide, tau, mitochondrial respiratory chain, adenine nucleotide translocator, VDAC

## Abstract

Alzheimer’s disease (AD), certainly the most widespread proteinopathy, has as classical neuropathological hallmarks, two groups of protein aggregates: senile plaques and neurofibrillary tangles. However, the research interest is rapidly gaining ground in a better understanding of other pathological features, first, of all the mitochondrial dysfunctions. Several pieces of evidence support the hypothesis that abnormal mitochondrial function may trigger aberrant processing of amyloid progenitor protein or tau and thus neurodegeneration. Here, our aim is to emphasize the role played by two ‘bioenergetic’ proteins inserted in the mitochondrial membranes, inner and outer, respectively, that is, the adenine nucleotide translocator (ANT) and the voltage-dependent anion channel (VDAC), in the progression of AD. To perform this, we will magnify the ANT and VDAC defects, which are measurable hallmarks of mitochondrial dysfunction, and collect all the existing information on their interaction with toxic Alzheimer’s proteins. The pathological convergence of tau and amyloid β-peptide (Aβ) on mitochondria may finally explain why the therapeutic strategies used against the toxic forms of Aβ or tau have not given promising results separately. Furthermore, the crucial role of ANT-1 and VDAC impairment in the onset/progression of AD opens a window for new therapeutic strategies aimed at preserving/improving mitochondrial function, which is suspected to be the driving force leading to plaque and tangle deposition in AD.

## 1. Introduction

More than 100 years after its discovery, there is still no cure for Alzheimer’s disease (AD), the most common neurodegenerative disorder, the etiopathology of which remains largely unclear, probably due to the highly complex and multifactorial nature of the disease and also likely because the model systems currently developed to elucidate the underlying pathological mechanisms of neuronal death occurring in AD recapitulate the disease only indirectly. Therefore, only a combined and translational analysis of as many experimental models as possible—cellular, animal, and human—can provide a picture that comes as close to the pathophysiological status of AD. What is currently certain is that researching the causes of this devastating disease remains one of the most complex challenges for those involved in the neuroscience field.

AD begins subtly and progresses relentlessly for many years, robbing the victim of memory, personality, and ultimately, life. It is characterized by memory and cognitive impairment, and its classic neuropathological hallmarks are two groups of protein aggregates: senile plaques and neurofibrillary tangles (NFTs) [1].

The propensity to aggregate is an intrinsic property of many proteins that have a tendency to change from a functionally competent native state to an aggregate state that resists the degradation reactions that the cell sets in motion in a ‘quality control’ process. The formation in the brain of these very stable aggregates can produce neurodegeneration, loss of cognitive functions, and more. Nobel laureate Linus Carl Pauling stated in the 1950s that proteins are ‘sticky’, although in native conditions, they are almost always soluble in water. Biological evolution has put in place a series of biochemical mechanisms that normally prevent aggregation through intracellular degradation or interaction with other molecules that mask the ‘sticky’ areas.

Among the progressive neurodegenerative proteinopathies, that is, diseases caused by the conformational change in a protein that normally has other roles in cell biology, AD is certainly the most widespread. In AD, the initial stages of the aggregation process are often slow and potentially reversible, therefore vulnerable; but the larger aggregates, stabilized by many intermolecular bonds, are essentially irreversible and end up forming the so-called amyloid plaques—containing mainly glial cells, dystrophic neurites, and amyloid β-peptide (Aβ) aggregates. Aβ, a peptide produced during the degradation process of the amyloid progenitor protein (APP), is present in two different types (Aβ 1–40; Aβ 1–42), depending on the number of amino acids from which it is composed (40 or 42) and the neural toxicity (Aβ 1–42 is considered the most toxic [2]). NFTs are also intracellular aggregates of coupled helical-shaped filaments whose main constituent is the hyperphosphorylated form of the protein tau (P-tau) [3]. The phosphorylation and dephosphorylation of the tau protein, mainly present in the axonal compartments of neurons, are physiological processes that regulate the binding of tau to microtubules, their assembly, stabilization, and axonal transport [4] of nutrients, vesicles, mitochondria, and chromosomes. Pathological tau is produced by perverse post-translational modifications, including phosphorylation, acetylation, ubiquitination, and truncation, which determine a lowering of its affinity with axonal microtubules [4], leading to its detachment from them and subsequent aggregation, formation of NFT, and synaptic dysfunction in AD [4]. It is precisely the P-tau form, together with the cleaved one, which contributes in an essential way to the beginning and progression of AD [1]. Although phosphorylation is generally considered to be one of the most important modifications of tau in AD [5], the tau protein is also a substrate of several endogenous proteases [6], and among these, caspases—activation is related to the toxic effects mediated by the Aβ [7]—and calpain are the most intensively studied [6]. In this context, it was found that tau can be cleaved into different residues of the carboxy-terminal region [7], preferably by cleavage of caspase-3 to aspartic acid 421 [7]. Another tau truncation occurs in the N-terminal domain by the action of caspase-6 [6,7]. Fragments of N-terminal tau have been found in various in vitro and in vivo models, such as in primary neuronal cultures undergoing apoptosis [7], in the cerebrospinal fluid of rats after head injury, in transient forebrain ischemia [8], and in the brain tissue of AD patients [9]. Further reports have demonstrated that a significant proportion of 20–22 kDa N-terminal tau fragments (NH_2_htau) are preferentially found in the mitochondria-rich synapses of the hippocampus and frontal cortex of AD subjects. Furthermore, the NH_2_htau fragment is associated with neurofibrillar degeneration and synaptic damage in the human AD brain [10].

The two toxic species, Aβ and tau, differ in order of both appearance and aggressiveness. Indeed, if the accumulation of Aβ in the brain of a person with AD is largely completed in a preclinical phase, known as a mild neurocognitive disorder, the accumulation of tau continues throughout the course of the disease so that, starting from parts of the brain called the entorhinal cortex and hippocampus, the total amount of abnormal tau in the brain of AD is linked to the stage and severity of the disease. Therefore, researchers attribute Aβ accumulation as a marker for early-stage disease and P-tau as a marker for late-stage disease [11]. For about 30 years, Aβ peptides have been considered one of the most relevant therapeutic targets for AD. However, the idea that Aβ pathology is not responsible for the onset of the disease has taken hold for some time [11], after the disappointing failure of many clinical studies based on the hypothesis that the Aβ cascade is not free from weaknesses, such as the striking example in which elderly patients, with a high load of amyloid plaques, do not develop significant cognitive deficits, while other patients, even with less accumulations, show severe symptoms of dementia.

Nowadays, the certain idea is that AD is not the direct neuropathological consequence occurring downstream of the deposition of Aβ alone, followed by the appearance of modified tau, but rather a multifactorial disease [12], including mitochondrial dysfunction as one of the most well-accepted and popular hallmark [12]. Additionally, the same AD-related neuropathological players (APP, Aβ, and tau) interact with mitochondria and alter mitochondrial function [13], although there is also evidence to support the hypothesis that the same abnormal mitochondrial function can trigger neurodegeneration and aberrant processing of APP or tau. A change of perspective is certainly necessary, that is, shifting attention to the early stages of disease, trying to identify the molecular mechanisms and phenomena that occur upstream of the formation of Aβ, that is, when full-blown dementia is still absent.

This is why strategies targeting mitochondria as disease modifiers in neurodegenerative proteinopathies are a coveted field of research.

In this regard, we cannot fail to mention a very latest study by Yoshida et al. [14] in which the authors, using transgenic AD mice (5XFAD) at different ages and isolated mitochondria, measured reactive oxygen species (ROS)-related protein expression and remarked that mitochondrial dysfunction through oxidation is strongly correlated with the onset of AD. Further, several research groups have found mitochondrial functional defects in 3XTg-AD transgenic mice [15] and in P301L, P301S, and PS2/tau transgenic mice [15]. On the other hand, oxidative damage is a discovery that cannot be overlooked because patients’ brains develop oxidative damage years before the accumulation of Aβ and tau aggregates [16]. This finding is also confirmed in animal and cellular models of AD showing increased oxidative damage prior to amyloid deposition [17,18]. Relevantly, an intriguing theory, as it links together different aspects of AD pathology (mitochondrial dysfunction, oxidative stress, and protein aggregation), reveals that the same Aβ peptide has an antioxidant effect against the oxidative stress induced by mitochondria [19,20], with Aβ1–40 more than other isoforms, including Aβ1–42. Aβ has also been shown to cause ROS production [21,22], and both β-secretase activity and tau hyperphosphorylation are increased by the action of ROS [23,24], underlining that mitochondrial ROS per se can exacerbate the accumulation of Aβ and tau aggregates.

The purpose of this review is to highlight the role played in the progression of AD of two ‘bioenergetic’ mitochondrial proteins, that is, the adenine nucleotide translocator (ANT) and the voltage-dependent anion channel (VDAC): they are transport proteins inserted in the mitochondrial membranes, inner and outer, respectively.

However, before going into the merits of the matter, we will devote ourselves to the description of a ‘must’, that is, how mitochondria work and what happens in them in AD—just to introduce/accompany even the less accustomed reader—so as to be able to understand the importance of continuing research on mitochondria–pathoprotein interaction. Considering that, from the movement of a finger to the creation of a memory, the actions of the human body require a harmonious concert of protein interactions; discovering how proteins interact with each other is fundamental to understanding ‘what is wrong’ in disease states, such as AD, and, in turn, developing new prevention and treatment strategies.

## 2. Mitochondria—What You Need to Know to Understand the Rest!

Mitochondria are fascinating organelles that combine the food we eat and the air we breathe to generate energy so that every other function of our body can take place. Four compartments are distinguished in the mitochondria, namely, the outer membrane (MOM), the intermembrane space (IMS), the inner membrane (MIM), and the matrix, with interdependent functions in the harmony of cellular energy metabolism. Until a few years ago, the MOM, characterized by a high lipid/protein ratio, was considered freely permeable to low-molecular-weight substrates; today it is known that its permeability is regulated in physiological conditions by modulating the activity of VDAC. The IMS is the mitochondrial compartment between the MOM and the MIM. This last is formed by phospholipids, with a high presence of cardiolipin (CL), a molecule characteristic of this structure, where it plays a fundamental role in a large number of processes (we will see it further below). In the lipid bilayer of the MIM, the proteins—some with specific transport functions, others engaged in the production of energy—are localized in an anisotropic way, an indispensable requirement for processes, such as oxidative phosphorylation, (OXPHOS) to take place. The matrix, the internal space of the mitochondrion, contains a large number of enzymes involved in various metabolic pathways: the urea cycle, the production of the heme group, the Krebs cycle, and the oxidation of fatty acids.

The scientific vulgate has it that mitochondria are ‘the energy powerhouse of the cell’, a definition that today is a little reductive but substantially correct. In fact, it is in these organelles that energy is produced in form of ATP through OXPHOS, which involves proteins organized in five complex structures and which together constitute the respiratory chain (RC). In these structures, electrons from the oxidation of carbohydrates and fats are collected by specialized molecules, NAD^+^ and FAD, and used to accumulate protons (H^+^) in the IMS; finally, they are transferred to oxygen with the formation of water. Protons accumulated in the space between the membranes are used to form ATP in the same way in which the water of a dam is used to produce electricity: the cascade of protons that flow back towards the matrix supplies energy to a protein, FO-F1 ATP synthase, which is a real turbine capable of using the electrochemical gradient, that is, the different concentration of protons between the IMS and the mitochondrial matrix, to produce almost all the ATP needed by the cell. The mechanism is prodigious in its refinement, complexity, and efficiency, yet it is not without its problems. To be honest, in fact, mitochondria do a very dangerous job: they handle oxygen, a very reactive substance, and electrons, particles charged with energy. The passage of electrons through the mitochondrial structures is subject to a more or less significant reduction in the amount of energy produced. In addition, it has far more important consequences when it comes to aging: it leads to the formation of substances whose name alone inspires fear and tremor: free oxygen radicals! Of all the oxygen used in cellular respiration and therefore in the production of energy, 1%–2% produces free ROS. This jeopardizes the mitochondrial membranes, mtDNA, CL, and much more. In reality, there are ‘defense’ systems, but when the amount of ROS becomes too high, it becomes a serious problem for our health. In this complex scenario, ROS are not always the bad guys of the situation: in aged and inefficient mitochondria, it is precisely the accumulation of these reactive species that signals the start of autophagy processes, allowing the elimination of nonfunctional organelles (mitophagy), together with proteins and other damaged molecules. In the most extreme cases, however, when the impairment of cellular structures is advanced, it is still the mitochondria that trigger the processes that lead to apoptosis. This process is essential to regulate the development and growth processes of tissues and also to eliminate cancer cells, but it can become a problem when it is activated improperly, leading to the destruction of cells of the nervous system, as occurs in some neurodegenerative diseases and, unfortunately, even just as we get older.

All of the above means that we would not literally be alive without the mitochondria and makes us reflect on the fact that the goal of our survival is to make friends with dear old mitochondria to survive for a long time.

Ordinary people continue to know little or nothing about mitochondria, but their importance has been recognized by research for their effects on aging, apoptosis, metabolism, and many diseases. Numerous scientific works show that mitochondrial dysfunctions occur in pathologies, affecting nerve tissues even before clinical symptoms. They seem to play a crucial role in triggering degenerative processes, and the same dysfunctions seem to unite several neurodegenerative diseases, confirming the hypothesis that mitochondria and their early alterations are the ‘common thread’ that unites many neurodegenerative diseases, including AD.

## 3. Proteinopathy and Mitochondrial Dysfunction in AD: A Marriage

Although it is assumed that ‘Ready-Steady-Go’ in AD involves a stochastic folding of a nontoxic Aβ peptide into a toxic one, which in turn triggers downstream a stochastic folding of the tau protein, it is not to be underestimated that the cellular environment, in which all this takes place, is not alien to what is going on. The most renowned hypothesis is that mitochondria play a critical role in cellular proteostasis, in view the fact that mitochondrial dysfunction seems to precede decline and promote the aggregation of pathogenic proteins in AD [25]. It is also certain that pathogenic proteins and defective mitophagy increase the accumulation of dysfunctional mitochondria [26], getting bogged down in a ‘vicious circle’. Mitochondria-associated proteostasis and the contribution of mitochondria to prevent proteostatic crisis in the cytosol have been also considered by Ruan et al. [27,28].

We are still wondering what is the chronological sequence of the appearance of dysfunctional mitochondria and pathogenic proteins. In animals that develop the typical symptoms of AD, mitochondrial abnormalities are observed in the embryonic stage and in juvenile mice, long before Aβ accumulation [29]. It has also been documented that tau and Aβ act synergistically to accelerate mitochondrial dysfunction in animal models [30,31,32], highlighting as well that the effects of tau on mitochondrial activity occur through the addition of Aβ [11,33]. Therefore, in terms of molecular pathogenesis, the proteinopathy, that is, the abnormal accumulation of misfolded proteins, and the mitochondrial dysfunction are pivotal characteristics that occur and concur together in AD, reinforcing each other to guide the progression of the disease in the brain.

The overall picture that emerges from studies is that these two molecules damage the mitochondria in multiple ways, marching separately and striking together. Both insults would cause increased ROS levels, decreased activities of antioxidant enzymes, and an altered mitochondrial membrane potential, resulting in decreased ATP production.

Considering the excellent reviews on the causal associations of Aβ and tau with mitochondrial dysfunctions in AD [34,35,36,37], we do not intend here to provide a compendium of the countless studies on the alterations of mitochondria, whose extensive loss aggravates the disease development, but we want make a quick overview of some of the literature data, the most significant, which will be useful for introducing the main topic we intend to treat.

However, before starting, we would like to make a premise that will be useful to the reader: the data on the subject all have an apparent conflicting nature. The reason for this is to be found in the fact that each study is conducted on a relatively small number of participants, uses different techniques for preserving brain samples for analysis, and analyzes different samples, such as peripheral blood cells from living AD patients or postmortem brains, rather than animal models and even worms! The lack of models that replicate the entire spectrum of the clinical symptoms associated with AD is certainly the biggest obstacle that research in the field has always encountered. It is also important to consider that the changes observed in post-mortem studies may be a consequence of the evolution of the disease and may not be present at the beginning of the disease itself; therefore, these alterations are potentially unsuitable to become candidates as biomarkers and/or be targeted by therapeutic molecules.

Using biochemical, molecular, and electron microscopy studies on post-mortem AD brains or transgenic mouse brains, several groups have investigated the connection between Aβ and mitochondria and between tau and mitochondria. Aβ is associated with mitochondria [13,38,39] and is transported to the MIM through the translocase of the machinery of the MOM (i.e., TOM) [40]; pathological forms of tau are also localized in the mitochondria [7,41], interfering with mitochondrial function in different brain areas through multiple mechanisms [30].

Several studies have confirmed that APP and Aβ are critically involved in mitochondrial dysfunction and neuronal damage responsible for the progression of AD [42]; in particular, Aβ binds to mitochondrial proteins, such as the mitochondrial fission protein dynamin-related protein 1 (Drp1) [43], the mitochondrial protein of the MOM VDAC [15], the proteins of the mitochondrial matrix Aβ-binding alcohol dehydrogenase (ABAD), and cyclophilin D (CypD) (see [15]).

In a very recent study, Ring et al. [44] investigated the intracellular Aβ1–42-induced toxicity mechanisms using yeast and fly models expressing Aβ1–42 by identifying the HSP40 family member Ydj1 (DnaJA1 in humans) as a key player in Aβ1–42-triggered intracellular toxicity. The authors proposed that Ydj1/DnaJA1 drives mitochondria-dependent cell death through the stabilization of Aβ1–42 oligomers and their translocation into mitochondria.

Consistently, several studies suggest that both the P-tau and the cleavage form of tau significantly influence mitochondrial function and bioenergetics [1]. More importantly, both cleaved and hyperphosphorylated tau have been colocalized in the temporal cortex of the AD brain, suggesting that caspase-3 may contribute to the cleavage of P-tau (in Ser306/404) and then to neurodegeneration [45]. Therefore, truncated tau and P-tau could be actively present simultaneously, indicating that the presence of both pathological species could mediate tau-mediated toxicity by interacting with several membrane-bound mitochondrial proteins, including ATP synthase, creatine kinase U-type mitochondrial, and Drp1 [46]. Furthermore, P-tau increases the level of Drp1, which leads to an increase in the enzymatic activity of GTPase and excessive mitochondrial fragmentation [47]. Manczak and Reddy [15] found that P-tau also interacts with VDAC1, causing mitochondrial dysfunction in the AD brain.

Moreover, the N-terminal fragment of tau is also associated with mitochondrial membranes [42]: Amadoro et al. [48] reported that the 20–22 kDa NH_2_-derived tau fragment was extensively enriched in human mitochondria from AD brain synaptosomes and that the amount of tau correlated with pathological synaptic changes and resulted in functional impairment of organelles. Furthermore, Atlante et al. [49] investigated the relationship between the overexpressed N-terminal tau fragment (26–44 aa) and mitochondrial dysfunction, reporting that cytochrome *c* oxidase (COX) and ANT are main targets for the N-terminal tau fragment, but that ANT is the only mitochondrial player responsible for the impairment of OXPHOS and the low cellular availability of ATP.

Evidence from animal and cellular models and human post-mortem brain of patients with sporadic AD suggests that both Aβ and tau have a destructive effect on the electron transport chain (ETC). In this regard, the study conducted by Cardoso et al. is particularly interesting: the authors added Aβ to the medium of Ntera2 (NT2) cells and ρ0 derivative cells, that is, NT2 cells previously depleted of their mitochondrial DNA (mtDNA) and therefore missing the respiratory chain subunits. Aβ treatment did not damage the ρ0 cells, suggesting that mitochondria, especially ETC, mediate Aβ toxicity. The same is true of the tau protein, which localizes in mitochondria and interferes with mitochondrial function through multiple mechanisms [50].

Unfortunately—and this, to be honest, is part of the dynamics of scientific research—studies investigating the expression of OXPHOS proteins have revealed potentially conflicting results. Then, if a microarray analysis of post-mortem frozen hippocampal specimens has revealed a global decrease in nuclear encoded OXPHOS protein subunits and no change in mitochondrial DNA-encoded subunits when AD brains are compared with both aged-matched controls and patients with mild cognitive impairment [51], it is equally true that, on the contrary, not only both total cellular mtDNA and complex IV protein levels in the hippocampus and frontal and temporal lobes of AD increased in AD [52], but also they are found outside the mitochondria, suggesting an increased turnover of mitochondria or a decrease in their proteolytic breakdown [52]. Again, the expression of the mitochondria-encoded subunits of complexes III and IV increases in the AD brain, while those of complexes I and V (i.e., ATP synthase) decrease [53]. Proteomic and functional studies found that the expressions and activities of mitochondrial complexes I and IV were significantly deregulated [54]. In this regard, experiments of quantitative proteomics carried out on the AD triple transgenic mouse model (pR5/APPsw/PS2 N141I) revealed that the deregulation of the activity of complex I was tau dependent, while that of complex IV was Aβ dependent.

These data complement those obtained in another triple transgenic mouse model, 3XTg-AD (P301Ltau/APPSw/PS1 M146L) for refs see [3], in which mitochondrial dysfunction was evidenced by an age-related decrease in the activity of the system’s regulatory enzymes of OXPHOS, such as COX, or of the Krebs cycle, such as pyruvate dehydrogenase [55]. It is important to stress out that these alterations in the mitochondrial proteome of the cerebral cortices of 3XTg-AD mice occurred well before the development of significant Aβ and NFT plaques, supporting the idea that mitochondrial dysregulation is an early event in the pathogenesis of AD.

The reasons for the severe involvement of COX are not fully understood, but according to biochemical experiments conducted on the isolated enzyme treated with Aβ1–42, some authors have suggested that a direct inhibition of complex IV may depend on the dimeric binding of Aβ and on the reduction of Cu^2+^ to the active site of the enzyme [56]. Moreover, it has been proven that heme-a, the prosthetic group essential for the transfer of electrons for complex IV, is deficient due to the overabundant binding of Aβ to regulatory heme [57]. A direct physical interaction between Aβ1–42 and the mtDNA-encoded subunit 1 of complex IV also appears to be responsible for the enzymatic defect [58]. It is certain that, although the reduction of COX activity and expression [56,59] is not reflected in hypothetical mechanisms of alteration, the structure of the complex IV protein represents the most documented change in AD, possibly due to oxidative stress [60]. In fact, it is thought that it is due to a lipid peroxidation of the membrane since its activity is strictly CL dependent. However, we have questioned some previous observations on the possible role of COX in AD. Indeed, although it has been proposed that a decrease in COX activity may be the cause of mitochondrial dysfunction, we show that the inhibition of COX activity by the tau fragment NH_2_-26–44 is of secondary importance since COX does not restrict electron flow along the respiratory chain; that is, the capacity of COX exceeds that required to support respiration.

Mitochondrial ATP synthase, which uses the proton driving force through the MIM to bind ADP and Pi into ATP, is also defective in AD (see above). In this connection, a physical binding between the protein subunit of ATP synthase, which confers sensitivity to oligomycin (OSCP) and Aβ, was ascertained in human brains with AD and in transgenic mice. This interaction causes a decrease in OSCP and loss of ATP synthase function, with a subsequent decrease in ATP synthesis and an increase in ROS [61].

However, in addition to complex IV and ATP synthase, other molecular targets of Aβ alter mitochondrial function. For example, another Aβ-linked mitochondrial protein is ABAD. It is not part of OXPHOS, but the conformational change induced by the Aβ–ABAD interaction has been observed to cause mitochondrial dysfunction [62]; consistently, the inhibition of the ABAD–Aβ interaction partially prevents Aβ-induced mitochondrial dysfunction [63].

These data support the hypothesis that the turnover of mitochondria is altered in the brain of AD, with the potential and obvious consequence of a reduction in ATP levels in AD.

The destruction of the ETC is accompanied by alterations of various enzymes involved in the Krebs cycle, such as pyruvate dehydrogenase, α-ketoglutarate dehydrogenase (αKGDH), isocitrate dehydrogenase, and malate dehydrogenase [53], thus compromising the maintenance of mtΔΨ and, therefore, as anticipated, the mitochondrial production of ATP [61]. Reduced αKGDH activity has been observed, as well as in brain tissue, in peripheral cells of patients with AD [38], thus suggesting that it is possible that mitochondrial dysfunction in AD patients occurs not only in the central nervous system but also in peripheral tissue cells [38].

Understanding the nature of the reciprocal relationship between these damage mechanisms will be useful in identifying neuroprotective strategies in therapy. This is why this research should be promoted. Many researchers enter this matter with a straight leg by advancing the hypothesis that the restoration of mitochondrial function could increase neuronal survival and perhaps provide a therapeutic tactic to reverse the course of AD disease [64]. Welcome!

## 4. ANT and VDAC Both Interact with AD Toxic Proteins

Here, we gather all the existing information on the interaction of ANT and VDAC with toxic AD proteins, not providing an extensive compendium of the numerous studies in the literature, but rather conveying the intriguing and, often, controversial nature of their evidence. We will magnify the defects of ANT and VDAC, two distinctly ‘bioenergetic’ mitochondrial proteins, which are measurable hallmarks of mitochondrial dysfunction.

(*i*)ANT hosts two (opposite) functions, both involved in the control and regulation of cell fate [65,66]: one vital function, the other lethal. The vital role of ANT is the enzymatic one, which is the historical and the main function to catalyze the carrier-mediated exchange between the cytosolic ADP with the ATP formed in the matrix, by facilitating the export of the newly synthesized ATP into the cell and, at the same time, by providing ADP as a substrate available for its mitochondrial phosphorylation to ATP by ATP synthase. This function has been extensively characterized in mitochondria isolated from various tissues with radiolabelled nucleotides and in native ANT-containing proteoliposomes [67,68,69], but not only [70], as we will see! The lethal function of ANT, which occurs in conditions of cellular ‘sickness’, is associated with its involvement as a component of the mitochondrial permeability transition pore (mPTP), a structure that forms in the inner mitochondrial membrane and is thought to underlie regulated cell death [71,72].(ii)VDAC1 is the most abundant protein in the MOM since its discovery in 1976 [73]. Although its localization remains predominantly mitochondrial (mVDAC), VDAC has been found on the plasma membrane (plVDAC) of cells. Besides, although plVDAC has been extensively studied, its exact biological function is not yet known, and we will not address it here. Therefore, in this review, we will refer to mitochondrial VDAC as VDAC. Previously, VDAC was considered responsible for the near-free permeability of the MOM [73] or a large mesh sieve [74], but nowadays, it unexpectedly stands as a gatekeeper for the entry and exit of mitochondrial metabolites, thus controlling the cross-talk between the mitochondria and the rest of the cellular compartments [75,76]. Not surprisingly, therefore, VDAC has been implicated in a wide range of pathologies associated with mitochondria [77,78,79]. It is believed that the uniqueness of this channel derives from its key position at the interface between the mitochondrion and the cytosol [77,78,79,80], thus becoming an important hub protein that interacts with over 200 proteins [81,82,83] that regulate the permeability of the MOM. A plethora of cytosolic proteins, glycolytic, such as hexokinase, and other enzymes [78], as well as their neighbors in the MOM, such as the cholesterol transporter, [78] as well as proteins of the family Bcl-2 [78], interact with VDAC. An acute observation by Reina and De Pinto [84] pointed out to the world that the high diversity of natural and synthetic ligands with which VDAC interacts is a symptom of lack of specificity for VDAC, so that this protein is not used to being considered a reliable drug target.

### 4.1. ANT

Taking into account that, in apoptosis, the ATP, produced in the mitochondria, reaches the cytosol, through ANT, where it is used in a variety of processes, including the activation of caspase [85], it follows that the role of ANT in apoptosis—as well as in neurodegenerative diseases in which apoptosis is likely the aberrant process that leads to cell death—is crucial.

In a study about 15 years ago, we measured the activity of ANT in cultures of cerebellar granule cells (CGCs) undergoing apoptosis. In this regard, we briefly describe this model of apoptosis here: it mimics AD-like molecular events, such as the activation of the amyloidogenic process, the cleavage of tau, and the production of toxic fragments [85] and references therein], events that are caused by the induction of apoptosis. In the development of this AD model, we have identified that the time course of apoptosis occurs in two phases: in the first, that is, the early phase (0–3 h), production of ROS occurs—modulation of which is carried out by the antioxidant system (AOS), in turn modulated by proteasomes [86]—and the release of cytochrome *c* (cyt *c*) from the mitochondria occurs; in the second, that is, the late phase (3–8 h), there is cyt *c*-dependent caspase activation and degradation of the AOS due to both proteasome and caspase action, as does caspase-dependent degradation of the released cyt *c* [87,88].

In order to investigate ANT activity, we conducted a very sophisticated experiment, in which we simultaneously measured the ADP/ATP exchange and the opening of mPTP as a function of apoptosis time, in the absence or presence of compounds designed to selectively influence ROS production, such as the antioxidant and proteolytic systems, in order to determine whether these parameters were related to each other. A double alteration of the ANT occurs due to the action of ROS, in the initial phase, and of caspase, in the late phase of apoptosis. The loss of ANT transport function and the opening of mPTP are mutually dependent and are inversely correlated in late—but not early—apoptosis, whereby the progressive decrease in the transport function of ANT occurs simultaneously with the opening of mPTP. In this regard, although mitochondrial research in defining the molecular mechanisms of mPTP has puzzled for almost 70 years, now it is certain that the mPTP acts as a regulator of mitochondrial ion homeostasis and the effector mechanism of cell death [72]. In addition, the long-standing hypothesis that it originates ‘also’ from ANT is re-evaluated.

It is worth noting that mPTP opening appeared to depend on the action of caspase on components of mPTP other than the ADP/ATP translocator. Indeed, although a progressive decrease in ANT transport function can be prevented [89] by the caspase inhibitor z-VAD, the carrier itself is not prey of caspase because the ANT protein has not changed in quantity or molecular weight during apoptosis. In addition, none of the amino acid sequences of the three ANT isoforms recovered from the Swiss-Prot Data Bank were found to contain a consensus sequence for proteolysis by the caspases themselves.

The study of ANT—as a dented ‘bioenergetic’ protein that contributes to mitochondrial dysfunction in AD—came back to the fore when Professor Calissano’s group aimed to decode the functional role of the N-terminal domain of tau—as opposed to the tail C-terminal containing microtubule binding domains. The authors showed that the overexpression of human tau and some of its N-terminal fragments leads to cell death by necrosis, mediated by the N-methyl-d receptor aspartate (NMDAR) and caspase independently [90]. After that, we investigated whether and how chemically synthesized NH_2_-derived tau peptides (i.e., NH_2_-26–44 and NH_2_-1–25 fragments) affect mitochondrial function: OXPHOS was significantly reduced by the tau fragment NH_2_-26–44, but not by 1–25, with two privileged targets, COX and ANT, but only the latter turned out to be the target responsible for OXPHOS impairment. This finding explains the mechanism of action of the toxic tau fragment NH_2_-26–44, which is supposed to bind ANT and, as a consequence, cause the reduced availability of ATP leading to the release of glutamate from the cell and excitotoxic death from excessive NMDAR stimulation in AD [91].

An aggravated mitochondrial impairment has been described in triple APP/PS/tau transgenic mice carrying plaques and tangles, in comparison with mice that overexpress either protein, namely, the tau fragment or the APP. In particular, the tau fragment was found to interact preferably with the Aβ peptides in human AD synapses in association with ANT1 and CypD. The two peptides, Aβ 1–42 and NH_2_-26–44, the smaller and more toxic one, inhibit ANT-1-dependent ADP/ATP exchange and together further aggravate mitochondrial dysfunction by exacerbating ANT-1 impairment. Certainly, these data are in line with results obtained from histological analyses conducted in the AD brain and in mouse models, which point out the synergy between plaques and tangles [91,92,93,94]. In order to identify the plausible mechanistic interaction of truncated Aβ and NH_2-_tau peptides in impaired neuronal mitochondria, data on reciprocal coimmunoprecipitations on synaptic-enriched mitochondrial fractions suggest that an NH_2-_tau/ANT-1/Aβ/CypD complex exists within an authentic cellular context in AD but not in age-matched nondemented controls. At this point, we only had to investigate the potential clinical relevance of NH_2-_26–44 tau and Aβ 1–42 on ANT-1 activity. To perform this, having established that each peptide individually altered the activity of ANT-1, we investigated whether and how the decrease in ADP/ATP exchange induced by each peptide could vary under two different conditions: (1) when the addition of one preceded that of the other and (2) when peptides were added together with the system. The experimental procedure adopted is always the same—to which we have even dedicated a paper [70], to which the reader is referred—the one that allows the continuous monitoring of ATP efflux from neuronal mitochondria incubated with ADP [70]. Several observations were made:(i)The rate of appearance of ATP was reduced when Aβ1–42 or NH_2-_26–44-tau was added before ADP.(ii)The extent of inhibition was less than that found when the NH_2_-tau fragment was added alone if the incubation of homogenate with Aβ1–42 preceded the addition of NH_2-_26–44-tau, suggesting that the binding of Aβ1–42 to ANT1 resulted in a conformational change of the transporter protein, making it less accessible to the NH-tau fragment. This only happens if Aβ 1–42 is added before the tau fragment, but not vice versa, suggesting that Aβ 1–42 acts as a negative modulator of mitochondrial NH_2_-tau fragment toxicity and not vice versa;(iii)The extent of inhibition increased strongly when the two peptides, NH_2-_26–44-tau and fibrillar Aβ1–42, were added together, confirming that the two peptides together cooperate by potentiating ANT-1 dysfunction and further aggravating the production of ATP.

Amazed by these results, in an attempt to understand the process that modulates the interaction of the NH_2_htau and Aβ1–42 peptides with ANT-1, we took advantage of an experimental model consisting of homogenates of cerebellar granule cells (CGCs) added with AD peptides, as the most suitable experimental system to obtain information both on the target protein of Aβ1–42 and NH_2_htau (i.e., mitochondrial ANT-1) and on the interaction between the carrier protein and the AD peptides themselves.

Since the binding of ADP to its active site involves thiol groups [95] and becomes susceptible to increased ROS production in AD neurodegeneration [70], a reversible alkylating agent of thiol groups oriented towards the external hydrophilic phase, mersalyl, was used in our experiments to selectively block and thus protect, in a reversible way, the –SH groups of ANT-1. Interestingly, we found out that the inhibition of ANT-1 by NH_2_htau and Aβ1–42 was due to an interaction between these two AD peptides, ROS and the thiol groups of ANT (Figure 1). We also discovered that the superoxide anion, the production of which occurs at the level of complex I of the RC in the presence of Aβ1–42 [96] and whose release from the mitochondria is significantly reduced in the presence of 4,40-diisothiocyanostilbene-2,20-disulfonic acid (DIDS), a VDAC inhibitor, modifies the thiol group(s) present at the active site of mitochondrial ANT-1, alters ANT-1 in a manner prevented by mersalyl, and abrogates the toxic effect of NH_2_htau on ANT-1. In view of these results, we can speculate that the interaction of Aβ-NH_2_htau on ANT-1 in AD neurons involves the thiol redox state of ANT-1 and the increase in ROS induced by Aβ1–42. This finding suggests the possibility of using various strategies to protect cells at the mitochondrial level, stabilizing or restoring mitochondrial function or interfering with energy metabolism and, ultimately, providing a promising tool for the treatment or prevention of AD.

Furthermore, we sought to determine whether NH_2_htau can also influence the distribution, shape, and size of mitochondria. Thus, using both in vivo and in vitro systems, we found that pathological NH_2_htau improves the turnover of mitochondria by adversely affecting their fusion/fission dynamics and their clearance by selective autophagy [97].

In a study we conducted shortly thereafter, being aware of the protective capacity of ADP towards its translocator [98], we demonstrated that ADP extracellular, following its internalization in neuronal cells, prevents mitochondrial ANT-1 compromise [98] and saves cells from dying by apoptosis. At the same time, the ability of ADP to protect ANT-1 from the toxic action of the two AD peptides (i.e., Aβ1–42 and NH_2_htau) was also verified. Collectively, these data were found to be consistent with the results of others demonstrating a cytoprotective action by adenosine during cell injury and with the discovery that adenosine and ADP exert a specific and marked antiapoptotic action in cultures of CGCs [99]. Unfortunately, at present, it cannot yet be said which is the true neuroprotective molecule: whether ADP itself or one of its degradation products.

### 4.2. VDAC

After 40 years of functional and structural studies, entrusting it with the role of a large mesh sieve, today VDAC appears to be at the crossroads between metabolism and cell death. In this article, we examine the involvement of VDAC in AD, in light of very important findings according to which VDAC is involved in the interaction with cytosolic proteins prone to aggregation, including Aβ and tau, by directly contributing to the onset/progression of AD. Until more than 10 years ago, the role of VDAC in AD was not known until the observation that its opening and closing were impaired in the mitochondria of brain tissues of patients with neurodegenerative disease nailed down its cardinal role in the pathogenesis of the disease [100].

A growing body of evidence indicates that there is a direct involvement of VDAC in mediating mitochondrial toxicity related to misfolded aggregates, and in many cases, it even appears that VDAC1 is the preferred docking site at the mitochondrial level for these proteins. Dated research [101] revealed that in cortical tissues of APP transgenic mice, the protein levels of VDAC1 were significantly increased in 12- and 24-month-old mice compared with 6-month-old mice, indicating that an age-dependent increase of VDAC1 occurred in the cerebral cortex of these AD mice in a similar manner of affected postmortem brains. Following studies on transgenic doubles, APP/PS1, and triples, 3XTg-AD [15] matched with the observation that Aβ and phosphorylated tau were found to be associated with mitochondrial membranes, causing mitochondrial dysfunction. In particular, in APP mice and AD brain tissues, VDAC1 is very abundant around senile plaques and NFTs [102]. The first exploratory steps in this regard revealed that VDAC colocalizes with full-length APP and oligomeric Aβ in the human frontal cortex of patients with AD, thus confirming its relevance in the progression of AD [15]. These histological studies involved the use of Aβ antibodies (6E10 monoclonal A11 oligomeric Aβ) and VDAC1 antibodies [42]. Similar to monomeric Aβ, oligomeric Aβ was also found in immunoprecipitation eluates from patients with severe AD and from APP and APP/PS1 mice. Taken together, these studies clearly suggest that both monomeric and oligomeric Aβ interact with VDAC1 and block pores with further damage to mitochondria in AD neurons. In support of a tight correlation between Aβ levels and VDAC1 expression, it is interesting to underline that soluble Aβ oligomers were able to induce VDAC1 upregulation in a human neuroblastoma [15] cell line.

That the interplay between Aβ and VDAC1 occurs was proven by adding soluble Aβ oligomers to SH-SY5Y cell cultures: increased levels of total VDAC1 was observed, and in addition, VDAC1 phosphorylation facilitated the escape of mitochondrial proapoptotic molecules contributing to the neurotoxic effects of Aβ [103].

Double-labeling immunofluorescence analyses of VDAC1 and P-tau were conducted to ascertain whether P-tau also interacts with VDAC1 in the brains of AD patients and control subjects and in the brain tissues of transgenic and wild-type mice [42].

The involvement of VDAC in direct interaction, not only with Aβ [15,104], but also with P-tau, contributes to impaired opening and closing of mitochondrial pores [15], to the interruption of the transport of mitochondrial proteins and metabolites between the mitochondria and the rest of the cell, to the impairment of the conductance of the channel itself (i.e., VDAC). This combination of interactions between VDAC1 and Aβ and between VDAC1 and P-tau leads to imbalances in metabolite fluxes across the MOM, resulting in mitochondrial dysfunction in AD neurons. Based on all these observations, lowering the levels of VDAC1, Aβ, and P-tau could reduce the interactions between VDAC1 and Aβ and between VDAC1 and phosphorylated tau in AD neurons, resulting in the maintenance of normal opening and closing of mitochondrial pores and then in the normal functioning of mitochondria supplying ATP to nerve endings (as is the case with healthy neurons). If these hypotheses were valid, the reduced interaction between VDAC1 and the two toxic proteins could prove useful for enhancing synaptic and cognitive functions in AD [42]. The key question, which Manczak’s group attempted to answer with a very interesting study, is how much VDAC1 is minimally sufficient to reduce the extent of the interaction between VDAC1 and Aβ and between VDAC1 and P-tau in order to maintain synaptic activity, mitochondrial function, and neuronal survival in the brains of transgenic mice with AD [47]. The authors, starting from the observation that homozygote VDAC1 knockout (VDAC1−/−) mice exhibited disrupted learning and synaptic plasticity, whereas VDAC1+/− mice appeared normal in terms of life span, fertility, and viability relative to control mice, aimed to determine mitochondrial activity in VDAC1+/− mice and VDAC1+/+ mice by focusing their attention on the characterization of mitochondrial and synaptic genes. In summary, the authors found that VDAC1+/− mice, notable for their reduced VDAC1 levels compared with VDAC1+/+ mice, exhibited improved mitochondrial function and synaptic activity, and reduced the expression of several AD-related genes. Therefore, reduced expression of VDAC1, such as that in VDAC1+/− mice, may be beneficial for synaptic activity and may protect against the toxicity of AD-related genes, eventually leading to normal function of the mitochondria that supply ATP to nerve terminals and to the enhancement of synaptic and cognitive function in AD.

Of course, it is absolutely essential that further studies delve into this finding to further evaluate VDAC1 reduction as a possible therapeutic approach in people with AD.

In addition to being characterized by enhanced expression levels of VDAC1 [104], AD is also characterized by a reduced interaction of VDAC1 with glycolytic enzymes, such as HKs [105]. Recent experimental evidence has revealed that (a) the trafficking of VDAC1 between the mitochondria and the cytosol is modulated by HK1 and (b) the VDAC1/HK1 interaction, as well as their dissociation, regulates the passage of VDAC1 between an open and a closed conformation, thus regulating cell survival/death balance [106] Figure 2.

As it has been investigated and deepened by many facets, HK—the enzyme that catalyzes the first phase of glycolysis, the phosphorylation of glucose (GLU) into glucose-6-phosphate (G6P)—binding to VDAC [107], obtains direct access to the mitochondrial ATP pool for GLU phosphorylation [79]. HK-1, interacting with VDAC, induces the closure of VDAC in a way that is reversed by G6P [108]. To support the link, there is evidence of HK-1-VDAC coimmunoprecipitation shown by Shoshan-Barmatz et al. [79].

Agents known to detach HK from mitochondria also induce ATP depletion, which prevents the lowering of cell viability [79,109].

In particular, in a study conducted by Bobba et al. [107], a decrease in VDAC1 activity was demonstrated due to its interaction with HK1 in the initial phase of apoptosis, while, in the late phase of apoptosis, the accumulation of G6P, following the glycolytic slowdown, induces VDAC1-HK1 dissociation, the recovery of VDAC1 activity and, consequently, cell death [107].

In an attempt to understand the exact mechanism by which the association of HK1 with VDAC1 has a metabolic advantage by suppressing apoptosis, we started from the observation that, in AD (1), there is a metabolic shift from OXPHOS to glycolysis (the so-called Warburg effect) and (2) that glucose metabolism is enhanced with the upregulation of key proteins that they internalize and metabolize glucose itself, like HK-1, while there is a parallel decrease in oxygen consumption by the mitochondria [110]. Thus, in the early and late phase of apoptosis, we assessed the expression level and activity of VDAC1, before examining its interaction with HK-1. We were able to follow simultaneously the ADP/ATP exchange by measuring the activities of two proteins, namely, VDAC1 and ANT-1, thanks to an experimental strategy: the measurement of the ADP/ATP exchange was carried out in the absence or presence of compounds capable of selectively blocking VDAC1 in the presence of DIDS, or ANT-1 with ATR, so as to be sure that the exchange of ADPext with ATPint was actually mediated from these two proteins and occurred exclusively through the mitochondrial membranes, external and internal, both intact. Although we advise the reader to deepen the experimental procedure carried out [107], here, it is important to point out that, for the first time, the activity of VDAC1 was measured in conditions close to physiological ones, unlike previous studies performed by using artificial systems reconstituted with phospholipid double layers [111]. Furthermore, the variations in the activity of VDAC1—it abruptly decreases in the first phase and recovers in the late phase of apoptosis—are independent of the relative abundance of the protein since its amount remains unchanged, as verified by Western blotting analysis. Thus, these results suggest that VDAC1 regulation occurring in our in vitro conditions was not attributable to differences in intracellular expression levels but rather to a direct modulation of enzymatic activity. Conversely, as reported above, in the post-mortem brain of AD patients, VDAC1 and/or VDAC2 levels are significantly reduced or elevated in different brain regions [112], and VDAC1 is overexpressed in the hippocampus of an amyloidogenic transgenic mouse model [113,114]. Subsequently, in order to investigate whether—as happens in tumor cells [111]—HK-1 interacts with VDAC1 at the MOM, we also conducted coimmunoprecipitation studies by using an anti-HK-1 polyclonal antibody as bait antibody, followed by immunoblotting with a monoclonal antibody that specifically interacts with VDAC1. Coimmunoprecipitation results confirmed the presence of the VDAC1/HK-1 immunocomplex in apoptotic cells in the early phase. Surprisingly, in the late phase, the VDAC1 signal decreased dramatically, suggesting that the stability of the VDAC1/HK-1 interaction was inversely associated with the progression of apoptosis.

As far as the HK activity is concerned, we observed that it increased in the same time interval that VDAC decreased. Then, the high level of HK-1 [110] causes: (*i*) its physical binding to VDAC1, (*ii*) the stunning of mitochondrial function, and consequently, (iii) the low oxidative stress in the cell. In the transition to the late phase, an inverse trend in activities was observed: the activity of VDAC1 was substantially restored, while HK-1 activity reduced. Further, when the VDAC1/HK-1 complex was destroyed, the activities of both enzymes tended towards control values, with the reopening of the channel and the restoration of VDAC1 function, responsible for the awakening of mitochondrial functionality, ending in cell death. Therefore, we deduced that VDAC1 activity strictly depends on the physical interaction with HK-1: it is low when HK-1, linked to VDAC1, acts as a gate, while it increases when the physical interaction with the HK-1 enzyme is destroyed.

Consistently, some reports have shown that the HK/VDAC1 interaction in tumors prevents apoptosis [77,79,111,115], while the disruption of the HK/VDAC1 bond by mutagenesis of key amino acids on VDAC significantly improves the induction of apoptosis [79,116]. Precisely, it is G6P, product of the HK reaction, that antagonizes the bond of HK with VDAC1 see [75], thus involving the opening of VDAC1, which, in turn, acts as a wake-up call to reactivate mitochondrial function, including the absorption of ADP, Pi, and respiratory substrates; OXPHOS; and release of ATP into the cytosol. Overall, both glucose phosphorylation and mitochondrial binding of HK with VDAC1 contribute to the protective effects of HK-1, consistent with [107], in inhibiting apoptosis. However, HK-1′s attempt to promote cell survival through the interaction and closure of VDAC1 proves unsuccessful, and in the late phase, a functional VDAC1 is required for the progression of apoptosis.

Incidentally and consistent with these data, the same Aβ, interacting with VDAC1, causes the detachment of the antiapoptotic protein, HK, with a consequent increase in the conductance of the channel and the release of cyt *c* [43,117]. Conversely, several studies have also shown that the interaction of VDAC1 with Aβ can lead to channel closure. In particular, VDAC1 has been proven to interact with phosphorylated tau, another key component in the pathogenesis of AD, and, together with Aβ, cause the channel blocking.

## 5. Conclusions

The situation described is compelling: it may indeed be worth considering that mitochondria actually initiate AD, becoming the driving force that leads to plaque and tangle deposition [31,118,119]. In this case, this hypothesis would be strongly in conflict with the popular, current, and most accredited one, named the amyloid cascade [120,121], which has strongly influenced research on AD for three decades despite strong dissent [122]. The pathological convergence of tau and Aβ on mitochondria finally explains why the strategies used against the toxic forms of Aβ or tau have not given promising results individually and suggest potential new pathways and targets for a combined therapeutic intervention. Furthermore, the crucial role of ANT-1 and VDAC impairment in the onset/progression of AD opens a window for new therapeutic strategies aimed at preserving/improving mitochondrial function and cognitive functions.

From this roundup of information—sometimes conflicting—it is clear that we are facing an ‘impasse’ that should spur researchers to continue investigating the links between mitochondria and the proteins involved in AD, because only a better understanding of these could potentially lead to the identification of clinically useful neuroprotective therapies.

The search for effective drugs and strategies for the treatment of AD is clearly linked to the identification of pathogenetic targets; therefore, future research efforts should be invested in (*i*) understanding the real chronology of events, (*ii*) correctly locating the mitochondrial dysfunction within this timeline, and (*iii*) establishing once and for all whether mitochondrial dysfunction is a primary cause or a secondary event. Only when these three key points are solved correctly will it be easier to intervene pharmacologically, and no more time and money will be wasted on useless therapeutic studies.

Interesting are the reflections made in a recent study by Bachurin [123], who believes that since mitochondria have a key and very early role in the development of neurodegenerative processes [124,125], the search for pharmacological agents capable of normalizing the functioning of these subcellular organelles of vital importance for nerve cells is certainly to be considered a promising approach to the design of effective neuroprotective drugs [126,127,128]. In his opinion, the possibility of developing a new generation of multitargeted agents endowed with the ability of normalizing the state of mitochondria and then of enhancing memory and cognitive function through their action on receptor and enzymatic systems responsible for the transmission of the synaptic signal deserves to be fully investigated [129,130,131].

## Figures and Tables

**Figure 1 ijms-23-07722-f001:**
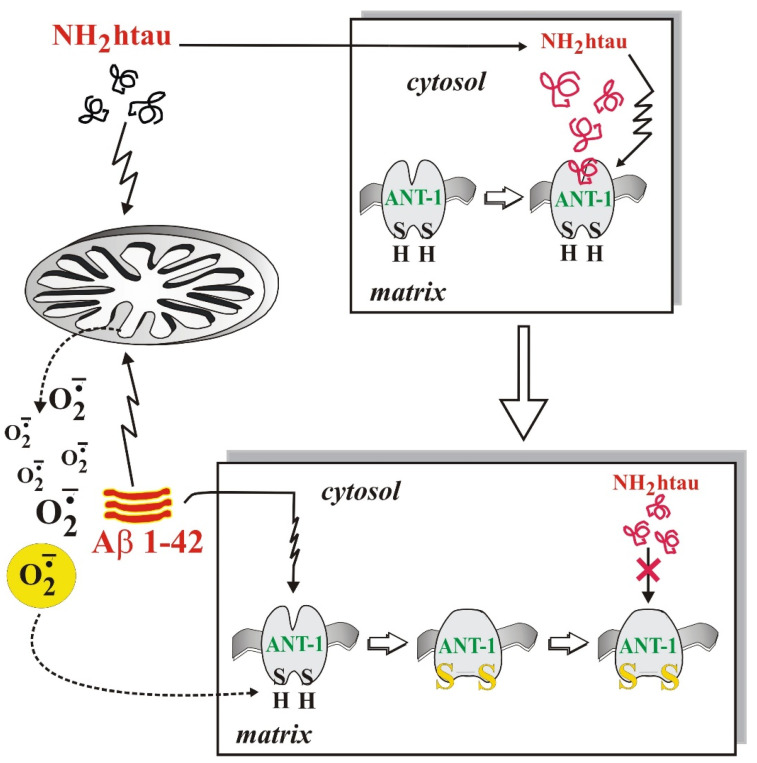
Both thiol groups present at the ANT active site and the Aβ1–42-induced ROS increase, which can oxidize these –SH residues underlying the pathological Aβ-NH_2_htau interplay on ANT-1 in AD. In the upper panel, the NH_2_htau fragment affects ANT-1 without interacting with Aβ. In the lower panel, Aβ induced-ROS production oxidizes ANT-1 thiol/s, thus modulating NH_2_htau toxicity.

**Figure 2 ijms-23-07722-f002:**
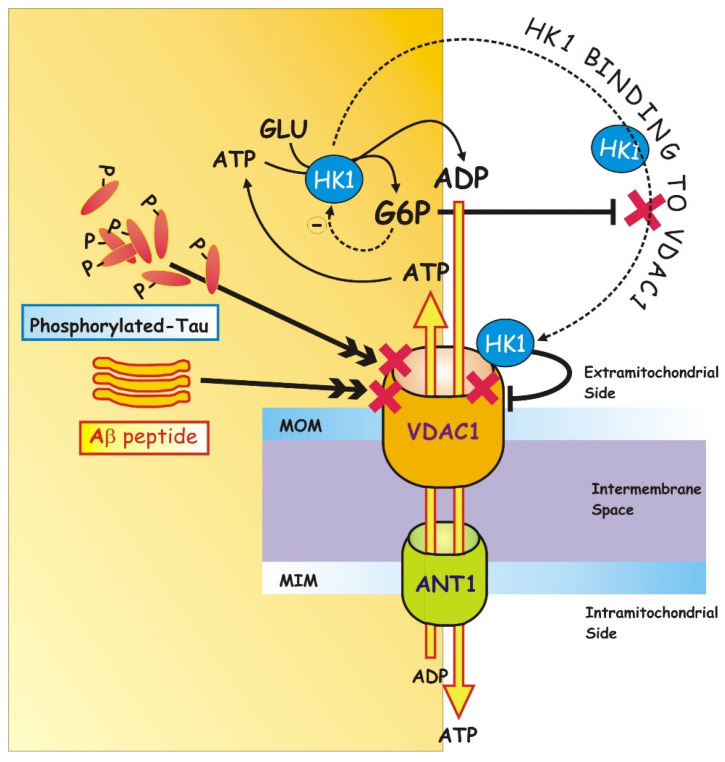
On the left, both monomeric and oligomeric Aβ interact with VDAC1 and block pores, as does P-tau [15]. This combination of interactions between VDAC1 and Aβ and between VDAC1 and P-tau leads to imbalances in metabolite fluxes across the MOM, resulting in mitochondrial dysfunction in AD neurons. On the right, the trafficking of VDAC1 between the mitochondria and the cytosol, modulated by HK-1, is depicted. HK-1, which catalyzes the phosphorylation of glucose (GLU) into glucose-6-phosphate (G6P), binds to VDAC1, thereby gaining direct access to the mitochondrial ATP pool for GLU phosphorylation. HK-1, interacting with VDAC, induces the closure of VDAC in an inverted way by G6P, capable of inducing the VDAC1/HK1 dissociation, the recovery of VDAC1 activity, and consequently cell death (for details, see the text).

## Data Availability

Not applicable.

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
