# Peer review of "Dysfunction of Mitochondria in Alzheimer’s Disease: ANT and VDAC Interact with Toxic Proteins and Aid to Determine the Fate of Brain Cells"

_ijms, 2022, doi:10.3390/ijms23147722_

Round 1

Reviewer 1 Report

The manuscript entitled "Dysfunction of Mitochondria in Alzheimer's Disease: ANT and VDAC interact with toxic proteins and aid to determine the fate of brain cells" by Atlante A. et al. is a very comprehensive and a very well written review, written by experts in the field. The manuscript is of great benefit for the broad readership of the journal. Therefore, I recommend publication of the manuscript after considering the following minor points.

Minor points:

1.) Part 3, page 5, lines 209 to 218: Mitochondria potentially contribute to cytosolic proteostasis to prevent proteostatic crisis in the cytosol (Ruan L et al., Nature 2017, PMID: 28241148). This and similar references should also be considered by the authors.

2.) The authors may consider to discuss novel findings described in Ring J et al., EMBO Mol Med 2022, PMID: 35373908.

Author Response

Reviewer 1

The manuscript entitled "Dysfunction of Mitochondria in Alzheimer's Disease: ANT and VDAC interact with toxic proteins and aid to determine the fate of brain cells" by Atlante A. et al. is a very comprehensive and a very well written review, written by experts in the field. The manuscript is of great benefit for the broad readership of the journal. Therefore, I recommend publication of the manuscript after considering the following minor points.

The authors thank the reviewer for her/his appreciation and positive comments on the manuscript.

Minor points:

1.) Part 3, page 5, lines 209 to 218: Mitochondria potentially contribute to cytosolic proteostasis to prevent proteostatic crisis in the cytosol (Ruan L et al., Nature 2017, PMID: 28241148). This and similar references should also be considered by the authors.

As suggested by the Reviewer, mitochondria-associated proteostasis and the contribution of mitochondria to prevent proteostatic crisis in the cytosol have been now considered and discussed by adding the related references of Ruan et al, 2017, 2020.

2.) The authors may consider to discuss novel findings described in Ring J et al., EMBO Mol Med 2022, PMID: 35373908.

The authors have inserted and discussed the novel findings described in Ring J et al., 2022

Reviewer 2 Report

The review by Atlante et al. is devoted to the analysis of literature data on the role of mitochondria in the development of Alzheimer's disease. In particular, the authors focused on the role of two proteins (adenine nucleotide translocator and voltage-dependent anion channel) in the progression of Alzheimer's disease. The authors suggested that a therapeutic strategy aimed at maintaining and improving the functioning of mitochondria could contribute to the suppression of the development of Alzheimer's disease.

This review is interesting and useful for the reader. As a remark, I would like to note that the authors, describing the role of ANT in the formation of MPT pores, refer to too old Refs [68, 69]. I would recommend the authors to replace these Refs with the following: PMID: 31238859, PMID: 34710270 containing updated data.

It is also not entirely correct to write that VDAC is “purely “bioenergetic” protein”. This protein exchanges not only ADP and ATP molecules between the cytoplasm and mitochondria, but also other metabolites and ions that are not directly involved in the bioenergetic functions of mitochondria.

Author Response

Reviewer 2

The review by Atlante et al. is devoted to the analysis of literature data on the role of mitochondria in the development of Alzheimer's disease. In particular, the authors focused on the role of two proteins (adenine nucleotide translocator and voltage-dependent anion channel) in the progression of Alzheimer's disease. The authors suggested that a therapeutic strategy aimed at maintaining and improving the functioning of mitochondria could contribute to the suppression of the development of Alzheimer's disease.

This review is interesting and useful for the reader. As a remark, I would like to note that the authors, describing the role of ANT in the formation of MPT pores, refer to too old Refs [68, 69]. I would recommend the authors to replace these Refs with the following: PMID: 31238859, PMID: 34710270 containing updated data.

First of all, we thank the Reviewer for her/his positive assessment of our review.

We agree with the reviewer for his/her observation and we have changed the references.

It is also not entirely correct to write that VDAC is “purely “bioenergetic” protein”. This protein exchanges not only ADP and ATP molecules between the cytoplasm and mitochondria, but also other metabolites and ions that are not directly involved in the bioenergetic functions of mitochondria.

Regarding the Reviewer's observation, perhaps due to the fact that our studies have always focused on bioenergetics, we are inclined to apply the term ‘bioenergetic’ to everything concerning the mitochondrion, in particular whether the two membranes, which play the main mitochondrial functions - i.e. the translocation of metabolites, the formation of the proton gradient, the synthesis of energy and so on - are involved. The outer mitochondrial membrane, the interface between the cytosol and the mitochondrion, is a limiting boundary for modulating cell bioenergetics, mediated via VDAC1. However, VDACs serve a myriad of functions ranging from energy and metabolite exchange to highly debatable roles in apoptosis. Their role in molecular transport puts them on the center stage as communicators between cytoplasmic and mitochondrial signaling events.

That said, if the Reviewer agrees, we would leave the term "bioenergetic proteins", referring to ANT and VDAC proteins, but delete the term 'purely'.
